# scientific report

# Tpo1-mediated spermine and spermidine export controls cell cycle delay and times antioxidant protein expression during the oxidative stress response

*Antje Krüger*[1,2], *Jakob Vowinckel*[1], *Michael Mülleder*[1], *Phillip Grote*[2], *Floriana Capuano*[1], *Katharina Bluemlein*[1†] *& Markus Ralser*[1,3+]

[1]Cambridge Systems Biology Centre and Department of Biochemistry, University of Cambridge, Cambridge, UK, [2]Max Planck Institute for Molecular Genetics, Berlin, Germany and [3]Division of Physiology and Metabolism, MRC National Institute for Medical Research, London, UK

**Cells counteract oxidative stress by altering metabolism, cell cycle and gene expression. However, the mechanisms that coordinate these adaptations are only marginally understood. Here we provide evidence that timing of these responses in yeast requires export of the polyamines spermidine and spermine. We show that during hydrogen peroxide ($H_2O_2$) exposure, the polyamine transporter Tpo1 controls spermidine and spermine concentrations and mediates induction of antioxidant proteins, including Hsp70, Hsp90, Hsp104 and Sod1. Moreover, Tpo1 determines a cell cycle delay during adaptation to increased oxidant levels, and affects $H_2O_2$ tolerance. Thus, central components of the stress response are timed through Tpo1-controlled polyamine export.**

Keywords: oxidative stress response; polyamines; cell cycle arrest; metabolite export; timing  

## INTRODUCTION

Oxidants such as hydrogen peroxide ($H_2O_2$) originate from metabolism as well as from environmental exposure, and are required in redox reactions and signalling [1,2]. However, when their concentrations exceed normal physiological levels they damage cellular macromolecules and cause oxidative stress. This pathological condition is considered a 'hallmark' of cancer and aging, and contributes to related pathologies [3–5]. On the other hand, the potentially harmful oxidative stress also bears yet unexplored therapeutic potential. Oxidant production increases with high metabolic activity. Thus, rapidly proliferating cells, for example, cancer cells or infective bacterial cells, have to compensate increased oxidant amounts, rendering them sensitive to pro-oxidant therapies [6,7].

Cells react to increased oxidant levels by arresting in the cell cycle, adjusting metabolism and through induction of antioxidant proteins. Most of our knowledge regarding the regulation of this response derives from studies of stress-responsive transcription factors [8,9]. These are activated through redox sensitive signalling cascades or cysteine oxidation, and induce expression of enzymes and molecular chaperones that support oxidant tolerance, such as superoxide dismutase, Hsp70, Hsp90 and Hsp104 [8–11]. Further gene expression changes are effectuated by altered metabolic activity [12–14]. The metabolic network is a prime target of the antioxidant machinery, as it produces both oxidizing and reducing metabolites, and thus changes in metabolism directly influence the redox balance. Indeed, during oxidative stress, a flux redirection from glycolysis to the NADPH-generating pentose phosphate pathway is established to compensate for the increased need of this cofactor by the antioxidant machinery [15–17]. As this redirection is rapidly induced by enzyme oxidation, post-translational modifications and metabolic feedback loops, it facilitates an immediate protection of the oxidant-exposed cell [13,18]. Despite the importance of this mechanism, it is still however unknown to which extent secondary metabolic fluxes contribute to achieve stress resistance.

Here, we report evidence that the timing of the stress response relies on a new metabolic rheostat control mechanism. We show that $H_2O_2$-exposed yeast cells export the polyamine metabolites spermidine and spermine via their transporter, Tpo1. This metabolite export times the induction of stress response proteins, including Hsp70, Hsp90, Hsp104 and Sod1, mediates overall $H_2O_2$ tolerance and prolongs the $H_2O_2$-induced cell cycle arrest. Hence, spermidine and spermine

[1]Cambridge Systems Biology Centre and Department of Biochemistry, University of Cambridge, 80 Tennis Court Road, CB2 1GA Cambridge, UK  
[2]Max Planck Institute for Molecular Genetics, Ihnestr. 73, 14195 Berlin, Germany  
[3]Division of Physiology and Metabolism, MRC National Institute for Medical Research, The Ridgeway, Mill Hill, London NW7 1AA, UK  
[†]Present address: Quotient Bioresearch, CB7 5WW Fordham, Cambridgeshire, UK  
[+]Corresponding author. Tel: +44 (0)1223 761346; Fax: +44 (0)1223 766002;  
E-mail: mr559@cam.ac.uk

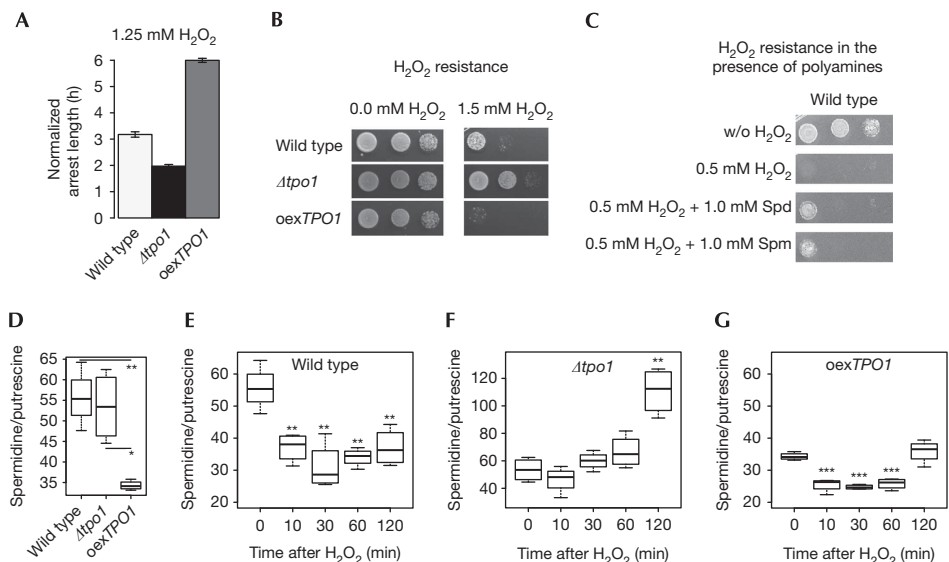

**Fig 1 | Tpo1 exports spermidine during the oxidative stress response. (A)** The $H_2O_2$-induced growth arrest is shortened in $\Delta tpo1$ cells and prolonged in *TPO1* overexpressing (oex*TPO1*) yeast. Wild-type, $\Delta tpo1$ and oex*TPO1* cells were grown exponentially in synthetic complete media (SC), exposed to 1.25 mM $H_2O_2$. Growth arrest duration was determined by R/grofit [32] and plotted relative to the arrest of untreated cells. Error bars represent s.d. ($n = 4$). **(B)** $\Delta tpo1$ cells are $H_2O_2$ resistant, whereas oex*TPO1* cells are $H_2O_2$ sensitive. Exponentially growing strains were spotted in 10-fold dilutions on SC plates containing 1.5 mM $H_2O_2$ and incubated at 30 °C for 3 days. **(C)** Polyamine presence in the growth media increases $H_2O_2$ tolerance. Spot testing as in **(B)**, but with wild-type cells spotted on SC plates containing $H_2O_2$ with or without spermidine (Spd) or spermine (Spm). **(D)** *TPO1* overexpression decreases spermidine concentrations. Intracellular spermidine/putrescine level, as determined by LC-MS/MS in exponentially growing wild-type and *TPO1*-mutant cells. **(E–G)** The spermidine concentration during the stress response depends on *TPO1*. Wild-type and *TPO1*-mutant cells were grown exponentially in SC, treated with 1.5 mM $H_2O_2$ and sampled in a time course. Error bars represent s.d. ($n = 4$); Student's *t*-test: $* = P \leq 0.05$, $** = P \leq 0.01$, $*** = P \leq 0.001$. **(E)** Spermidine levels in wild-type cells decline upon a $H_2O_2$ treatment. Spermidine/putrescine ratio in $H_2O_2$-treated wild-type yeast. **(F)** *TPO1* deletion reverses the spermidine trend and leads to spermidine accumulation. As in **E**, but with $\Delta tpo1$ yeast. **(G)** *TPO1* overexpression reduces spermidine levels during the stress response. As in **(E)**, but with oex*TPO1* yeast. $H_2O_2$, hydrogen peroxide; oex*TPO1*, overexpressing *TPO1*; SC, synthetic complete; Spd, spermidine; Spm, spermine.

concentrations are altered to control the timing of central components of the oxidative stress response.

## RESULTS AND DISCUSSION
### Tpo1 exports polyamines during the stress response
Yeast exposed to sub-lethal $H_2O_2$ concentrations temporarily arrests in the cell cycle [19]. We used the duration of this growth arrest as readout to screen for timing regulators of the stress response. 5,150 gene deletion strains, equalling the 'nonessential' *Saccharomyces cerevisiae* genome [20], were exposed to 1.25 mM $H_2O_2$ and their recovery from the oxidant exposure was followed photometrically. Compared with wild-type cells, 15 strains re-entered growth at a different time. Extensive quality tests (supplementary Information online), confirmed a monogenetic trait in a strain deleted for the plasma membrane transporter gene *TPO1* (YLL028W). Upon $H_2O_2$ exposure, $\Delta tpo1$ cells recovered growth faster than wild-type cells (Fig 1A). Conversely, a strain overexpressing *TPO1* (oex*TPO1*), created by genomic integration of a second, *GPD1* promoter controlled *TPO1* copy, arrested for a longer period (Fig 1A). This growth phenotype correlated with stress resistance. $\Delta tpo1$ cells were more $H_2O_2$ resistant than wild-type cells, whereas oex*TPO1* cells were $H_2O_2$ sensitive (Fig 1B).

Tpo1 is a plasma membrane exporter for the polyamines, spermidine and spermine [21,22]. These metabolites were first

identified in seminal fluid [23], but are ubiquitous and highly concentrated growth factors [24,25]. Their detailed molecular function is still under debate; however, they influence a broad range of cellular processes, including translation, transcription and autophagy [24,26–28]. Moreover, extracellular spermidine exposure prolongs lifespan of several organisms, including yeast [27], and polyamines have anti-inflammatory and antioxidant properties [24,26].

To investigate a potential role of spermidine and spermine in the antioxidant response, we first determined their effect on yeast's stress tolerance by testing for survival of the yeast strains on agar plates containing $H_2O_2$. The presence of spermidine or spermine in the growth media increased $H_2O_2$ resistance (Fig 1C). Next, we quantified intracellular spermidine and spermine levels by liquid chromatography tandem mass spectrometry (LC-MS/MS). Consistent with previous results, demonstrating that spermidine and spermine are exported through Tpo1 [22], *TPO1* overexpression markedly lowered the basal concentrations of both metabolites, whereas *TPO1* deletion had no significant influence on their basal levels (Fig 1D; supplementary Fig S1A online for spermine, supplementary Fig S5A–C online for absolute levels). Remarkably, upon $H_2O_2$ exposure, spermidine and spermine concentrations changed in a Tpo1-dependent manner. In wild-type cells, an immediate and significant decline in spermidine (Fig 1E), and a

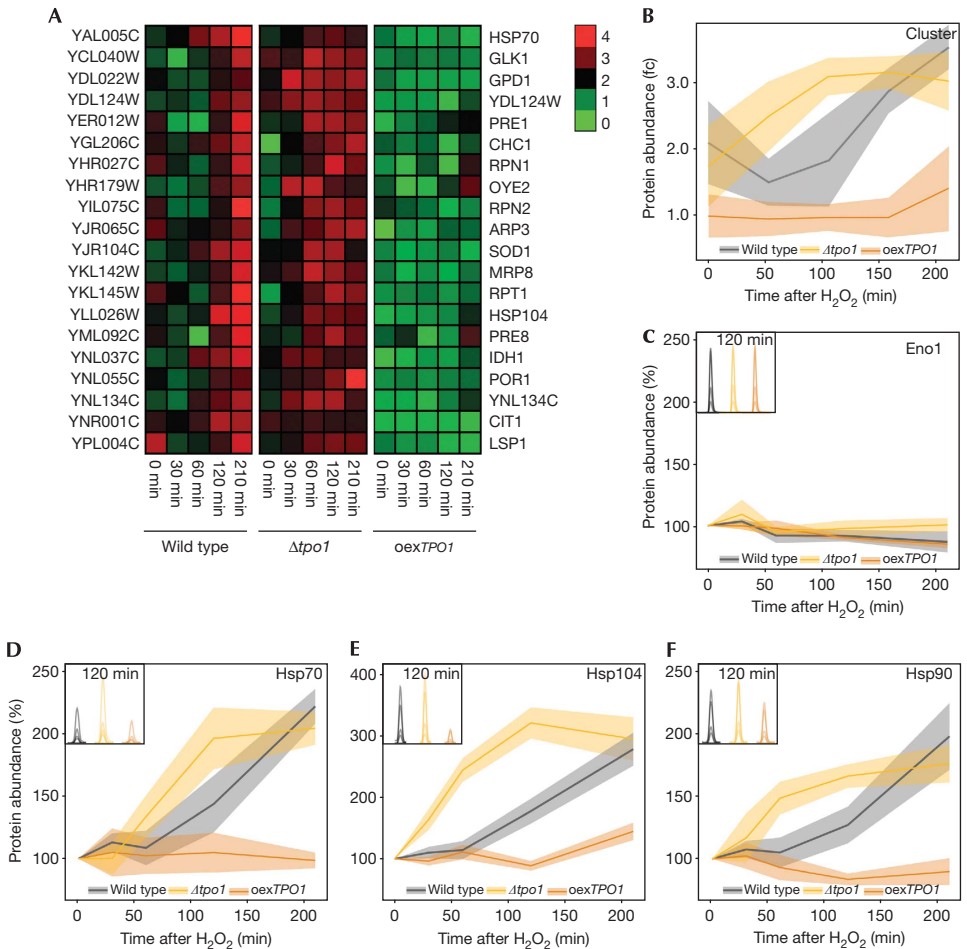

**Fig 2 | TPO1 times the activation of the stress response. (A–F)** Wild-type, Δtpo1 and oexTPO1 cells were grown exponentially, treated with 1.5 mM H₂O₂ and sampled at indicated time points. **(A)** Induction of stress response genes is accelerated in Δtpo1 cells, but prevented by TPO1 overexpression. Relative expression level of 404 proteins as determined by SWATH-MS [29] and expressed as fold change (0 = the median expression value of the individual protein). The heat map illustrates 20 proteins identified by co-clustering with Hsp104. **(B)** Polyamine export controls the timing of stress gene activation. Summary diagram of the relative expression of proteins identified in **A** as determined by targeted analysis of proteomic data. **(C–F)** Expression of heat shock proteins is accelerated in Δtpo1 yeast and delayed in oexTPO1 cells. Relative expression of Hsp70, Hsp104 and Hsp90 as determined by targeted SWATH-MS. Shown are relative changes in abundance of at least four peptides per protein, each monitored by three MS/MS transitions. Embedded diagrams show chromatograms obtained for a representative peptide (VNQIGTLSESIK (Eno1), TTPSFVAFTDTER (Hsp70), NPSDITQEEYNAFYK (Hsp90) and VIGATTNNEYR (Hsp104)) after 120 min, where wild-type and TPO1-mutant cells differed most. Eno1, enolase; H₂O₂, hydrogen peroxide; oexTPO1, overexpressing TPO1.

statistical trend of the less abundant spermine (supplementary Fig S1B online) was measured. TPO1 deletion reversed this phenotype; spermidine and spermine levels did not decline, but instead accumulated (Fig 1F; supplementary Fig S1C online). In contrast, in oexTPO1 cells, both polyamines were retained at lower levels and did not accumulate (Fig 1G; supplementary Fig S1D online).

### TPO1 times the induction of the stress response

As polyamine concentrations influence translation [24,28], we speculated that the time-dependent concentration changes of spermidine and spermine could be associated with the induction of the stress response. Using a state-of-the-art technique in quantitative proteomics, SWATH-MS [29], we determined the relative expression of 404 proteins in wild-type and TPO1 mutants

during the stress response. Proteome profiles were recorded at different time points upon H₂O₂ exposure, and similarity clustering on the basis of Pearson correlation was used to identify TPO1-dependent regulatory clusters (Fig 2A). This analysis revealed that Tpo1 controls the induction of proteins required for oxidant tolerance [8,10,11]. Hsp90 (genes HSP82/ HSC82 [10]), Hsp70 (SSA1 [30]), Hsp104 [11] and Sod1 [31] were induced in wild-type cells as previously reported. In Δtpo1 cells, the induction of these proteins occurred faster. Contrarily, in oexTPO1 cells, their induction was delayed or their expression level remained unchanged (Fig 2A). Co-clustering with Hsp104 further identified 18 antioxidant enzymes, ribosomal components, chaperones and nucleotide synthesis factors (Pnc1) that followed the same pattern (Fig 2A).

Targeted analysis of the SWATH-MS data confirmed these results. Spectral information for representative peptides (supplementary Table S1 online) was extracted from the SWATH profiles and their peak intensities were normalized to a reference protein, Tdh1. Individual time course data are illustrated for the chaperones Hsp70 (Fig 2D), Hsp104 (Fig 2E) and Hsp90 (Fig 2F). Induction of these proteins was accelerated in $\Delta tpo1$ cells, and delayed in oex$TPO1$ cells. A similar result was obtained for the sum of all proteins of the cluster (Fig 2B). In comparison, the expression of a representative control protein, enolase, was not influenced by Tpo1 or $H_2O_2$ treatment (Fig 2C). Hence, on $H_2O_2$ treatment, stress response protein induction is determined in a Tpo1-dependent manner.

## $TPO1$ extends the oxidant-induced cell cycle arrest

The growth retardation that follows a $H_2O_2$ treatment is the consequence of a G2 arrest in the cell cycle [19]. We therefore tested whether the accelerated growth of $\Delta tpo1$ cells (Fig 1A) is caused by a deficient cell cycle arrest. However, wild-type and $\Delta tpo1$ cells accumulated comparably in the G2 phase on a $H_2O_2$ exposure (Fig 3A), indicating that the arrest was fully established.

Instead, growth assays revealed that cells deleted for $TPO1$ or overexpressing differ in the duration of this arrest. We investigated the growth response of wild-type and $TPO1$ mutants by exposing the cells to incremental $H_2O_2$ levels. The arrest time was calculated from the growth curves using R/grofit, employing a model-free spline fit [32], and was expressed as the time from treatment until the maximum growth rate was re-established. Comparing arrest time and oxidant dose, we observed that wild-type cells abruptly extend the cell cycle arrest (for 64%) when $H_2O_2$ concentrations exceeded 0.75 mM (Fig 3B, left and middle panels). Both below and above this level, $H_2O_2$ level and arrest time correlated in a linear fashion, resulting in a bi-linear correlation (Fig 3B, right panel).

This adaptation to high $H_2O_2$ concentrations was absent in $\Delta tpo1$ cells (Fig 3C); $H_2O_2$ dose and cell cycle arrest duration remained in linear correlation (Fig 3C, right panel). The faster growth recapitulation of $\Delta tpo1$ cells (Fig 1A) is thus the consequence of a deficient arrest extension (Fig 3C, left and middle panels).

In contrast, $TPO1$ overexpression reversed this phenotype. A lower $H_2O_2$ level was sufficient to prolong the cell cycle arrest in oex$TPO1$ cells, and once induced, the arrest lasted longer (Fig 3D). Hence, $TPO1$ is required for the adaptation to high $H_2O_2$ levels; cells lacking this gene were deficient in extending the cell cycle arrest in the presence of increased oxidant levels.

To test the role of polyamines in the arrest extension, we added spermine after the $H_2O_2$ treatment. This treatment restored an $H_2O_2$-induced arrest extension in $\Delta tpo1$ cells (Fig 4A,B; middle panels). Nonetheless, the arrest remained shortened compared with wild-type cells and required a higher $H_2O_2$ level for induction. Notably, this treatment prolonged the arrest in wild-type and oex$TPO1$ cells (Fig 4A, left and right panels; Fig 4B). As oex$TPO1$ cells tolerate higher spermidine and spermine levels [22], but arrested longer on spermine addition, while the polyamine-sensitive $\Delta tpo1$ cells ([22]) recommenced growth faster, we concluded that the prolonged arrest is not the consequence of polyamine toxicity. Moreover, a partially complementation for the arrest extension in $\Delta tpo1$ cells was

observed in complex media (YPD), which among other differences to synthetic media is rich in both polyamines as well (supplementary Fig S3 online). Therefore, the duration of the $H_2O_2$-induced cell cycle arrest is adapted depending on the applied $H_2O_2$ concentration; and this adaptation necessitates the presence of Tpo1 or its substrates spermidine and spermine.

## CONCLUSION

Survival during stress conditions requires rapid cellular adaptation, achieved through the stress response machinery. Although important features of this machinery have been identified, its dynamic and multi-layer hierarchical regulation is still marginally understood [8,33].

Here, we report that the polyamine exporter Tpo1 controls the levels of spermidine and spermine during the oxidative stress response and is involved in the coordination of two central response features: the induction of antioxidant proteins, including Hsp70, Hsp90, Hsp104 and Sod1, and the duration of the $H_2O_2$-induced cell cycle arrest. Metabolic export is thus central for mounting a timed induction of the stress response. In this context, further antioxidative protection might arise from direct oxidant depletion, as spermidine and spermine can scavenge free radicals [26]. Indeed, we detected an *in vitro* $H_2O_2$ depletion in the presence of spermidine and spermine. However, this effect was only moderate and non-stoichiometric (supplementary Fig S4B online), indicating that the direct antioxidant properties of polyamines have an additional, but presumably minor role during the $H_2O_2$ response.

$\Delta tpo1$ recommenced growth faster than wild-type cells, increasing their fitness during oxidant exposure. However, it is conceivable that the same behaviour could be deleterious under other circumstances, that is, when a second exposure would follow shortly after the first one. The prolongation of the cell cycle arrest of wild-type cells could thus be the consequence of an adaptation to defeat a repeated or cycling oxidant exposure. In this context, despite protein induction and cell cycle arrest being Tpo1-dependent, they appear to be regulated by functionally distinct mechanisms. As $\Delta tpo1$ cells are polyamine sensitive [22], and as spermine/spermidine uptake is mainly catalysed by Sam3 and Dur3 [34], indicates that polyamine uptake continues in the absence of $TPO1$. It is worth speculating that low intracellular polyamine concentrations prevent the continuation of the cell cycle until the stress response is completed and polyamine levels restored. Furthermore, there is evidence for extracellular polyamine sensing, as spermidine effects on the cell cycle arrest extension are additive in both $\Delta tpo1$ and oex$TPO1$, despite the latter strain has lower intracellular polyamine levels and is more spermidine/spermine resistant.

The literature contains several evidence that antioxidant properties of spermidine and spermine are conserved across species. First, spermidine and spermine are highly concentrated in cell types that have a high demand on oxidant protection, such as sperm [35]. Second, tumour cells whose cell growth is limited by high oxidative loads [6] have higher survival chances when they produce large amounts of polyamines [25,36]. Third, extracellular spermidine treatment does not only extend lifespan in yeast, but also in worms and flies [27]. Moreover, rat neurons trigger polyamine export upon stimulation [37]. Hence, both antioxidant properties and triggered export of

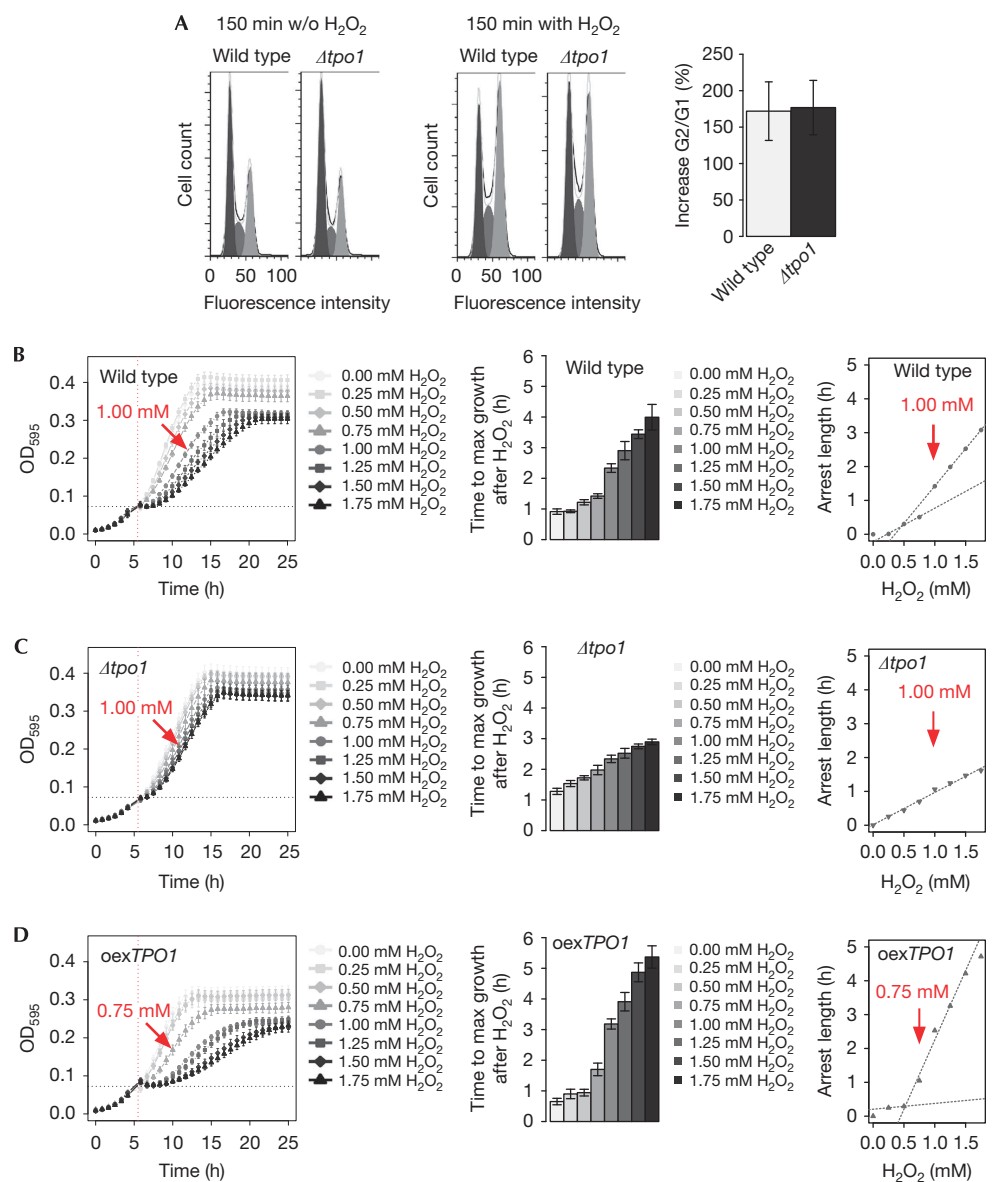

**Fig 3 | Tpo1 prolongs the $H_2O_2$-induced cell cycle arrest at high sub-lethal $H_2O_2$ levels. (A)** $H_2O_2$-treated wild-type and $\Delta tpo1$ cells arrest similarly in the G2 phase of the cell cycle. Wild-type and $\Delta tpo1$ cells were grown exponentially in YPD, then, treated with 0.75 mM $H_2O_2$ or left untreated for 150 min, sampled and stained with propidium iodide. Left and middle panels: Cell cycle distribution of at least 100,000 cells was measured by FACS and analysed using FlowJo 9.4.11 software. (Right panel) Relative increase in G2 over G1 cells. Error bars represent s.d. ($n = 3$). An analysis with 1.25 mM $H_2O_2$ is included in the supplementary Information online supplementary Fig S2 online. **(B–D)** Wild-type, $\Delta tpo1$ and *TPO1* overexpressing (oex*TPO1*) cells were grown exponentially in SC and treated with $H_2O_2$. Cell growth was measured photometrically and analysed with R/grofit. Error bars represent ± s.d. ($n = 4$). **(B)** Wild-type cells abruptly extend the $H_2O_2$-induced cell cycle arrest upon reaching a defined $H_2O_2$ concentration. Wild-type cells were treated with incremental $H_2O_2$ concentrations. Left panel: Growth curves as determined photometrically. Middle panel: Time until maximum growth rate is reached after treatment (*t*max). Right panel: Correlation of the $H_2O_2$-induced arrest length (*t*max minus basal value) and applied $H_2O_2$ concentration. Red arrow indicates the $H_2O_2$ threshold concentration inducing the prolonged cell cycle arrest. **(C)** $\Delta tpo1$ cells do not abruptly extend the cell cycle arrest at increased $H_2O_2$ levels. As in **B** but with $\Delta tpo1$ yeast. Arrest length and $H_2O_2$ dose form a single linear correlation (right panel). **(D)** *TPO1* overexpression prolongs the $H_2O_2$-induced cell cycle arrest. As in **B** but with oex*TPO1* yeast. $H_2O_2$, hydrogen peroxide; oex*TPO1*, overexpressing *TPO1*; SC, synthetic complete.

polyamines are observed in several organisms. It is now required to identify polyamine export systems in these species, and to test to which extent their stress response is dependent on polyamine export.

In conclusion, we identified a biochemical system that regulates the stress response through Tpo1-mediated export of the polyamine metabolites spermidine and spermine. This system appears to be central for achieving a time dependence in the coordination

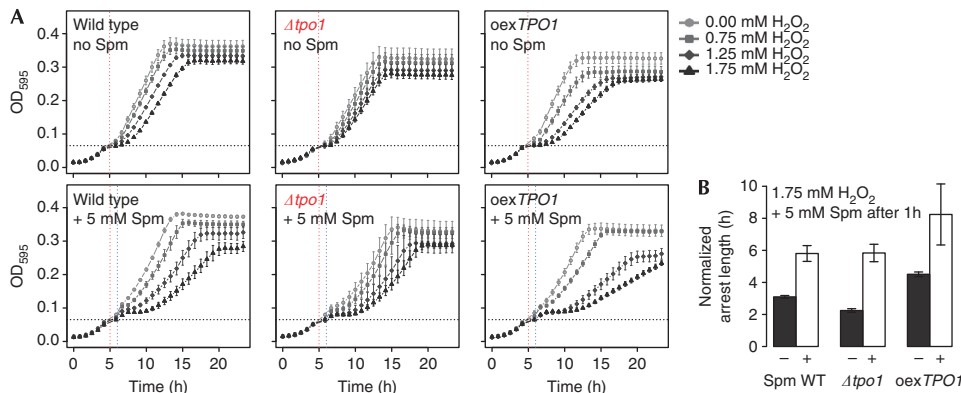

**Fig 4 | Extracellular spermine treatment restores the cell cycle arrest extension in Δtpo1 yeast. (A)** Wild-type, Δtpo1 (middle panel) and oexTPO1 cells (right panel) were grown exponentially in SC media, treated with $H_2O_2$ (upper panel) or with $H_2O_2$ and spermine 1 h after $H_2O_2$ addition (lower panel). Error bars represent s.d. ($n = 4$). **(B)** Arrest length of 1.75 mM $H_2O_2$-treated cells relative to non-$H_2O_2$-exposed cells when spermine was added 1 h after $H_2O_2$. Error bars represent s.d. ($n = 4$). $H_2O_2$, hydrogen peroxide; oexTPO1, overexpressing TPO1; SC, synthetic complete; Spm, spermine; WT, wild type.

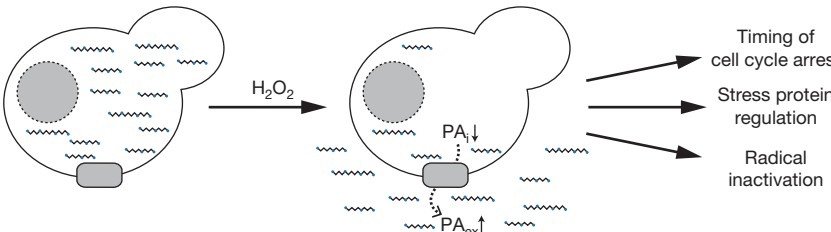

**Fig 5 | The role of TPO1-mediated spermidine and spermine export in timing the stress response. $H_2O_2$, hydrogen peroxide.**

of the stress response, affecting cell cycle progression and protein expression (Fig 5). Controlled metabolite export is thus a new regulatory principle in mediating the dynamics of the cellular stress response.

## METHODS

**Yeast strains.** All strains used are isogenic derivatives of BY4741 and listed in supplementary Table S2 online. Plasmid and yeast strain generation as well as yeast cultivation was conducted according to standard procedures as described previously [38].

**Screening the MATa gene deletion collection.** Four replicates of the 5150 strains [20] were grown in YPD to mid-log phase and exposed to 1.25 mM $H_2O_2$. Their growth was followed photometrically using a SpectraMax 250 Microplate reader (Molecular Devices).

**Growth assays.** Individual growth curves were determined using a multimode detector DTX 880 (Beckman Coulter) and analysed with R/grofit using a model-free spline fit [32].

**Flow cytometry.** Flow cytometry was performed on an AriaII SORP FACS (Becton Dickinson).

**Polyamine quantification.** Quantification of putrescine, spermidine and spermine was performed by LC-MS/MS using RP-HPLC (1290, Agilent) coupled to a Triple Quadrupole mass spectrometer, following derivatization with dansylic acid as described [39]. Spermidine and spermine concentrations were expressed relative to their precursor putrescine, which is not substrate of Tpo1 [22].

**SWATH-MS.** Samples were prepared according to our previous procedure [40] and analysed as described in Gillet et al, [29] on a 5600 QqTOF mass spectrometer (AB Sciex). Data were processed in Skyline [41] and Spectronaut (Biognosys).

### ACKNOWLEDGEMENTS
We thank our lab members for critical reading and discussing the manuscript, and Prof. H. Lehrach (MPIMG), S.G. Oliver (University of Cambridge), U. Stelzl (MPIMG), Z. Konthur (MPIMG) and F. Madeo (University of Graz) for support and materials. We acknowledge funding from the Max Planck Society, the Isaac Newton Trust, the Wellcome Trust (RG 093735/Z/10/Z) and the ERC (Starting grant 260809). M.R. is a Wellcome Trust Research Career Development and Wellcome-Beit prize fellow.

*Author contributions*: A.K., J.V., M.R. designed the study, analysed data and wrote the paper; A.K., J.V., M.M., P.G., F.C. and K.B. conducted experiments.

### CONFLICT OF INTEREST
The authors declare that they have no conflict of interest.

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
