## [Review Process File · EMBO Reports]

Manuscript EMBOR-2013-37930

Tpo1-mediated spermine and spermidine export controls cell cycle delay and times antioxidant protein expression during the oxidative stress response.

Antje Krueger, Jakob Vowinckel, Michael Muelleder, Phillip Grote, Floriana Capuano, Katharina Bluemlein and Markus Ralser

Corresponding author: Markus Ralser, University of Cambridge

Review timeline:

Submission date:	29 August 2013
Editorial Decision:	20 September 2013
Revision received:	20 September 2013
Accepted:	20 September 2013

Transaction Report:

Please note that the manuscript was previously reviewed at another journal and a revised version submitted to EMBO reports. The original reviews are not subject to EMBO's transparent review process policy and therefore cannot be published.

Editor: Esther Schnapp

1st Editorial Decision

20 September 2013

Thank you for the submission of your revised manuscript to EMBO reports. We have now received the comments from the referees who also reviewed the first version of this manuscript for another journal. Referee 1 still has a few minor suggestions, all of which I would like you to address in the manuscript text (especially the last point) before we can proceed with the official acceptance of your manuscript.

I look forward to seeing a new revised version of your manuscript as soon as possible.

REFeree REPORTS:

Referee #1:

I already like the first version of this interesting paper, and I am glad to say that the revised version addresses all of my concerns. Even though the authors do not completely unravel the molecular mechanism underlying the effect of polyamines on stress resistance, the observations described in the report are highly relevant to a broad group of scientists. I am also particularly happy with the discussion about polyamine uptake in a tpo1 deletion mutant, the possibility of an extracellular

sensor, and the improved readability of the introduction.

Minor remarks

Line 89 : "plasma membrane transporter TPO1" - replace with "the plasma membrane transporter gene TPO1"

Line 103 "spot testing" may not be known by non-experts; consider rephrasing.

Line 154 Replace "TPO1 mutants" by " Δ tpo1 mutants"

Line 159 Replace "9for 64%" by "(by 64%)"

Line 182-183: I am not convinced that the complementation in YPD provides any additional evidence for the involvement of polyamines - so many other factors could play a role here...

Comment #2 of reviewer 2 is, in my opinion, a very important comment. I think that the author's response in the rebuttal letter suffices to take away the concern. However, I would suggest including this discussion in the materials and methods or supplemental section.

Referee #2:

This work describes the interesting finding that deletion of the spermine/spermidine transporter Tpo1 increases the peroxide stress resistance of yeast cells, presumably by accelerating the induction of stress proteins. The results are very clear and the experiments are for the most part convincing. Although the study is mostly descriptive at this point and does not provide any mechanistic aspects how this increased oxidative stress resistance might be achieved in cells that lack this transporter, the overall findings are important and novel.

1st Revision - authors' response

20 September 2013

Thank you very much for delivering the positive reviewer comments. The minor suggestions of Reviewer #1 have been included.

To the YPD media: The reason why I would like to keep this figure as Supplementary material is as many yeast labs use YPD as their first choice of growth media. In case another labs will reproduce our results this will likely be the first choice of media as well, so it is important that they have the experiment for comparison (otherwise they might misinterpret the different growth phenotype between rich and minimal media as a conflicting finding)

In order to comply with the Reviewers comment, I changed the wording of the sentence in order to avoid any potential misinterpretation:

“Moreover, a partially complementation for the arrest extension in Δ tpo1 cells was observed in complex media (YPD), which among other differences to synthetic media is rich in both polyamines as well (supplementary Fig S3 online).”

2nd Editorial Decision

20 September 2013

I am very pleased to accept your manuscript for publication in the next available issue of EMBO reports. Thank you for your contribution to our journal.

As part of the EMBO publication's Transparent Editorial Process, EMBO reports publishes online a Review Process File to accompany accepted manuscripts. As you are aware, this File will be published in conjunction with your paper and will include the referee reports for EMBO reports, your point-by-point response and all pertinent correspondence relating to the manuscript.

If you do NOT want this File to be published, please inform the editorial office within 2 days, if you have not done so already, otherwise the File will be published by default [contact: emboreports@embo.org]. If you do opt out, the Review Process File link will point to the following statement: "No Review Process File is available with this article, as the authors have chosen not to make the review process public in this case."

Finally, we provide a short summary of published papers on our website to emphasize the major findings in the paper and their implications/applications for the non-specialist reader. To help us prepare this short, non-specialist text, we would be grateful if you could provide a simple 1-2 sentence summary of your article in reply to this email.

Thank you again for your contribution to EMBO reports and congratulations on a successful publication. Please consider us again in the future for your most exciting work.